# Optimization and Evaluation of Polymer Inclusion Membranes Based on PVC Containing Copoly-EDVB 4% as a Carrier for the Removal of Phenol Solutions

**DOI:** 10.3390/membranes12030295

**Published:** 2022-03-04

**Authors:** Agung Abadi Kiswandono, Candra Saka Nusantari, Rinawati Rinawati, Sutopo Hadi

**Affiliations:** Department of Chemistry, Universitas Lampung, Bandar Lampung 35145, Indonesia; candrasaka.n@gmail.com (C.S.N.); rinawati@fmipa.unila.ac.id (R.R.); sutopo.hadi@fmipa.unila.ac.id (S.H.)

**Keywords:** co-EDVB, DBE, membrane life-time, phenol, polymer inclusion membrane

## Abstract

Polymer inclusion membrane (PIM) is a method for separating liquid membranes into thin, stable, and flexible film forms. In this study, the PIM was made using polyvinyl chloride (PVC), dibenzyl ether (DBE), and 4% copoly-eugenol divinyl benzene (co-EDVB) as a supporting polymer, plasticizer, and carrier compound, respectively. Furthermore, a phenol transport test was carried out using the parameters of pH influence, the effect of NaOH concentration, and transport time. The PIM membrane was also evaluated using the parameters affecting the concentration of plasticizer, the effect of salt concentration, and the lifetime of the PIM membrane. The results show that the optimum pH obtained to transport phenol to the receiving phase was 5.5, with a concentration of 0.1 M of the NaOH receiving phase and a transport time of 72 h. Furthermore, it was found that the use of plasticizers and salts affected the ability and resistance of the membranes. The membrane lifetime increased up to 60 days with the addition of 0.1 M NaNO_3_ or NaCl salt in the source phase.

## 1. Introduction

The separation technology using liquid membranes has widely been accepted in many industries compared to other conventional methods such as liquid extraction, membrane filtration and electrodialysis [1]. PIM is the latest method that applies this technique using a composition of membrane components consisting of PVC as a basic polymer, and extractants commonly referred to as carriers [2]. It is one of the alternative methods that have been used in previous years due to its environmentally friendliness, which supports green chemistry. This is because PIM does not require large quantities of solvents but only relatively small amounts of carriers [3]. Furthermore, it has been widely used for both the separation of heavy metals [4,5,6,7,8,9,10], and organic compounds [11,12,13,14,15].

PIM demonstrates versatile capabilities with excellent stability when compared with other liquid membrane technologies. Another advantage is that it has a larger interfacial surface and is very selective depending on the type of constituents in order to obtain a membrane that is selective, strong, and able to easily and efficiently separate compounds [16]. The carrier compounds contained in the PIM method affect the lifetime of the membrane in transporting the target compound to the receiving phase. Furthermore, they also have a longer life span compared to that of SLM. This is because the mechanism of PIM transport depends on the composition of the membrane and its surface homogeneity [8].

The transport process in PIM requires a carrier compound as one of the components used in forming the membrane. Furthermore, there are various ways to improve the membrane stability and permeability such as adding a crosslinking polymer to the composition [3]. The interaction between the carrier and target compounds is also needed in order to transport phenol from the source phase to the membrane. The concentration of carrier compounds used in PIM is generally much smaller. This makes this method more economical to use than other conventional methods that use more solvents to increase the selectivity of the membrane [17].

In this study, 4% of copoly-eugenol divinyl benzene (co-EDVB) was used as a carrier compound because this compound has an OH active site that is suitable to facilitate the transfer of phenol pollutants by hydrogen interaction, and because co-EDVB is soluble in organic solutions [12]. Based on the physical and chemical characteristic of co-EDVB, it is chosen to transport phenol from the source phase to the receiving phase by evaluating the use of PIM. This was also in line with the study of Kiswandono et al. [12], whereby 2%, 6%, and 12% concentrations of co-EDVB were used in the phenol transport optimization test. A study on the ability of these carrier compounds to transport phenols under various conditions based on the lifetime of the membrane has also been carried out [18]. In this work, we reported the optimization and evaluation of PIM based on PVC containing co-EDVB 4%.

## 2. Materials and Methods

### 2.1. Materials and Transport Equipment

The instruments used in this work included a pH meter (Thermo ScientificTM Orion StarTM A211 Benchtop pH Meter, Ottawa, ON, Canada), FTIR (Cary 630, Agilent Technology, Santa Clara, CA, USA)), a UV-Vis 772 Spectrophotometer, (Shanghai, China), infrared spectrophotometer Shimadzu FT-IR 8201PC (Kyoto, Japan), a scanning electron microscope, SEM (JSM-6360LA, Tokyo, Japan), and a TG-TGA Perkin Elmer, (Billerica, MA, USA). The PIMs were installed between the two chambers for easy transportation (Figure 1). One chamber containing 50 mL of NaOH as a receiving phase and another chamber containing 50 mL phenol 60 ppm as a source phase. Then, the phenol concentrations contained in the source and receiving phases were analyzed using a UV-vis spectrophotometer at a maximum wavelength of 456 nm and the 4-aminoantiphyrine method [12]. The procedure was as follows: 5 mL of sample in the source phase, receiving phase and the phenol standard with several concentration variations were added with 5 mL of double distilled water so that the final volume was 10 mL. The pH of the solutions in the source phase and phenol standard was adjusted to 10 ± 0.2 using 1 M NH_4_OH, phosphate buffer, while pH for the receiving phase was adjusted with 0.5 M HCl, then 1 mL of 4-aminoantipyrine 2% and 8% potassium ferricyanide were added. The solution was allowed to stand until the color of the solution changed to pink.

After the color change occurred, the solution was transferred into a separating funnel and added with 5 mL of chloroform. The separating funnel was shaken and allowed to stand for a while until separation occurred, and then the organic layer or chloroform layer (bottom) was separated. The chloroform extract obtained was measured for absorbance using a UV-vis spectrophotometer at a wavelength (λ) of 456 nm.

The concentrations of the phenol in source phase and receiving phase were obtained from the calibration curve. Each measurement was repeated three times, and the observed %RSD data values (pH Variation: 0.45–5.64%; concentration of NaOH: 0.29–9.95%; time variation: 0.28–1.23%; variation of plasticizer: 0.27–2.87%) indicated that it was satisfactory over the range of concentrations.

### 2.2. Preparation of the PIM

The PIM consisted of three main components, namely 4% co-EDVB carrier compound, PVC as a base polymer, and DBE as a plasticizer. They were made using a total weight of 0.270, 0.5400, and 1.0800 g in a mold that has been equipped with a magnetic stirrer in the weight ratio of 10:32:58. In addition, 10 mL of tetrahydrofuran (THF) was used in each of the PIMs to homogenize the mixture in the mold which was allowed to stand for three days in order to evaporate the solvent naturally.

The installed PIMs were used for all phenol transport procedures are shown in Figure 1.

### 2.3. Phenol Transport

#### 2.3.1. Effect of pH in the Source Phase

The pH variation of the source phase was performed using a similar installed PIMs and was carried out by varying pH of phenol at 3.5, 4.5, 5.5, 6.5, and 8, while the concentration of NaOH as a receiving phase used was 0.5 M. Both chambers were stirred with a magnetic stirrer at room temperature for 9 h.

#### 2.3.2. Effect of NaOH Concentration

The effect of NaOH in the receiving phase was measured by varying the concentration of NaOH at 0.01, 0.05, 0.1, 0.25, and 0.5 M. The pH of phenol in the source phase was the pH optimum obtained in Section 2.3.1. Both chambers were stirred with a magnetic stirrer at room temperature for 9 h.

#### 2.3.3. Effect of Time

The effect of time on the phenol transport was analyzed using a similar PIM setting as above. The solutions were stirred with a magnetic stirrer at room temperature with time variations of 3, 5, 9, 12, 24, 48, and 72 h.

The ability of membranes to pass through a number of particles is called permeability. Furthermore, the permeability of a substance to pass through a membrane is expressed in a value called the flux value (J). Flux is the number of samples that pass through the membrane within a certain period of time. It is also referred to as the flow rate with respect to the thrust due to differences in concentration between the source and receiving phases. The membrane permeability coefficient value was found using Equation (1).
(1)$$P\left(\frac{m}{s}\right)=-\frac{V_{f}}{At}ln\frac{{\left[C\right]}_{t}}{{\left[C\right]}_{0}}$$
where *P* is the permeability (m/s); *V_f_* represents the sample volume in the source phase (L), [*C*]*_t_* is the sample concentration at time *t*; and [*C*]_0_ is the ample initial concentration s

The flux value can be obtained using Equation (2).
(2)$$J=\frac{V}{A\;t}$$

The flux value is known as *J* (L/m^2^ h), *V* represents the sample volume in the source phase (L), *A* is the membrane surface area (m^2^), and *t* represents time (hour). Furthermore, the flux value may also be used to determine the percent recovery of phenols in the sample.

#### 2.3.4. Transport Kinetics

Reaction order is part of the transport kinetics that is used to determine the transfer process of phenols from the source to the receiving phases. In addition, the first-order reaction model may also be determined from the coefficient value (R^2^) that is closest to the value using Equation (3).
(3)$$-ln\frac{{\left[C\right]}_{t}}{{\left[C\right]}_{0}}=\left(\frac{A\;k}{V}\right)t$$
where *k* is the coefficient of mass transfer.

### 2.4. Evaluation of the Ability and Resistance of PIMs

#### 2.4.1. Effect of Plasticizer Concentration

PIM was made with variations of plasticizer concentrations, namely 0.3032, 0.3100, 0.3132, 0.3200, 0.3232 g. Based on the procedure available in the literature [1], we adjusted the amount of plasticizer that we used so that the amount was appropriate so that the product was not too oily or stiff. Furthermore, the membrane was weighed before being transported and then placed between two chambers containing both the source and the receiving phases. Then, 50 mL of 60 ppm phenol and NaOH with optimum pH and concentration was added to the source and receiving phases of both chambers. The phenol solution was stirred for 48 h at room temperature under conditions not exposed to sunlight. Based on the chemical data sheet stating that safe storage conditions are in a cool place, the same treatment during the study was therefore carried out at room temperature and not exposed to sunlight. The concentration of phenol in both phases was measured with UV using the 4-aminoantiphyrine method [12].

#### 2.4.2. Effect of Salt Concentration

In the first procedure, the PIM with the optimum plasticizer concentration was placed between two chambers containing 50 mL of 60 ppm phenol at optimum pH mixed with NaNO_3_ salt. Concentrations of the NaNO_3_ salt, namely 0.0001, 0.001, 0.01, 0.1, and 1 M and 50 mL of NaOH were added into the solution in the source and receiving phases (without salt). The solution was stirred with a magnetic stirrer for 48 h at room temperature under conditions not exposed to sunlight. The concentration of phenol in both phases was measured with UV using the 4-aminoantiphyrine method.

In the second procedure, PIM with an optimum plasticizer concentration was placed between the two chambers containing 50 mL of an optimum solution of NaOH. This was mixed with NaNO_3_ salt with a concentration variation of 0.0001, 0.001, 0.01, 0.1, and 1 M in the receiving phase and 50 mL of 60 ppm phenol with optimum pH in the source phase (without salt). The solution was stirred with a magnetic stirrer for 48 h at room temperature under conditions not exposed to sunlight. In addition, the phenol concentrations contained in both phases were analyzed using a UV-vis spectrophotometer at a maximum wavelength of 456 nm and the 4-aminoantiphyrine method.

#### 2.4.3. Lifetime PIM Membrane

The PIM with an optimum plasticizer concentration was placed between the chambers of the two phases. The source phase received a mixture of two types of salt, namely 0.01 and 0.1 M NaNO_3_, and 0.01 and 0.1 M NaCl salts each at optimum pH. The receiving phase was also given a NaOH solution with an optimum concentration (without salt). Furthermore, the five kinds of transports were carried out at room temperature until the membrane became damaged. This was marked by changing the pH in the source phase to pH ± 9. It was found that when the pH changes, the phenol transport process was stopped.

## 3. Results and Discusion

### 3.1. Phenol Transport

#### 3.1.1. Effect of pH of the Source Phase

Benosmane et al. [11] and Pérez-Silva et al. [19] stated that the pH on the source phase affected the phenol transport, thus the variation on the pH was conducted at the initial stage in this work. The pH variations in the source phase were done at 3.5, 4.5, 5.5, 6.5, and 8 for transporting phenol.

The results of the transport process show that the phenol was transported at pH 5.5 with 33.74% content in the receiving phase (% Cp). It was also found that hydrogen bonding occurred and the π–π interaction between phenol and the carrier compound, namely co-EDVB, was on the membrane surface [12].

The results of the measurements using UV-vis spectroscopy show the amount of phenol that was transported from the source phase to the receiving phase as shown Figure 2. Furthermore, the percent removal of phenol that could have been transported was 47.04%, while the optimum phenol percent recovery was obtained at 33.7% in the receiving phase. Therefore, the more basic the pH of the source phase, the lesser the phenol content transported to the receiving phase.

#### 3.1.2. Effect of NaOH Concentration

The NaOH solution plays a major role in the receiving phase as a compound that attracts phenol in the membrane phase and converting it into a sodium phenolate compound.

Figure 2 shows the variations in the concentration of NaOH in the receiving phase, namely 0.01, 0.05, 0.1, 0.25, and 0.5 M.

The high concentration of NaOH proves that it influences the phenol content that is transported in the receiving phase. Based on Figure 3, the optimum phenol transport was obtained with a pH of 5.5 at 0.1 M NaOH concentration. Furthermore, the percent recovery of phenol transported in the receiving phase was 76.23%, while the phenol removal percentage was 88.8%. This phenomenon is coincident with the results reported by others [14,20].

#### 3.1.3. Influence of Transportation Time

Previously, Kiswandono et al. observed the effects of time on phenol transport efficiency and kinetics poly-bisphenol A diglycidyl ether (poly-BADGE). In this study, the effects of transport time on phenol transport efficiency were evaluated with copoly-EDVB 4% as the carrier [14]. The influence of time on phenol transport was also evaluated 3, 5, 9, 12, 24, 48 and 72 h. Figure 3 shows that this compound was transported at a higher rate as the time increases. Furthermore, the optimum time was 72 h with a transport value of 85.6%. This is because the interactions that occur between phenols in the membrane and receiving phases increased along with the time needed for transport.

Figure 4 shows that the transported phenol increases with increasing transport time. This is because the interactions that occur between phenol in the membrane and receiving phases increased along with the time needed for transport. The higher removal rate in the initial period can be attributed to the increase in a number of vacant sites on the carrier in the PIM [21].

### 3.2. Transport Kinetics

The ability of a membrane to allow a certain number of particles to pass through it is known as permeability. The permeability of a substance to pass through a membrane is expressed in a value called the flux value (*J*). In this paper, flux is defined as the number of samples that can pass through the membrane in a certain time, and can also be referred to as the flow rate against time due to the driving force due to the concentration difference between the source phase and the receiving phase [12].

Figure 5 shows a decrease in the flux value with increasing time, and the permeability of the membrane will also decrease to pass phenol from the source phase to the receiving phase (Table 1). Based on Equation (1), the membrane permeability coefficient value at the optimum time of 48 h was obtained at 8.21 × 10^−8^ m/s. Meanwhile, the flux at the optimum time of 48 h was obtained at 0.83 L/m^2^ h using Equation (2) with transported phenol recovery of 70.43%.

The reaction order is one part of the transport kinetics used to determine the process of phenol transfer from the source phase to the receiving phase. The reaction order model can be determined from the coefficient value (R^2^) which is closest to the value of 1. From the curve of 

$ln\frac{{\left[C\right]}_{t}}{{\left[C\right]}_{0}}$ vs. *t* in Table 2, the equation y = (−2 × 10^−6^)x + 0.7484 and the degree of slope (R^2^) = 0.989 were obtained. The first-order reaction model with a R^2^ value of 0.989 and a k value of 1.19 × 10^−6^ m/s is the most suitable in describing the transport kinetics in this study using Equation (3). The relationship curve between $ln\frac{{\left[C\right]}_{t}}{{\left[C\right]}_{0}}$ and the transport time can be seen in Figure 6. The first-order phenol transport kinetics curve in Figure 6 shows a decrease in the phenol concentration in the source phase corresponding to the first order.

### 3.3. Evaluation of the Ability and Resistance of PIM Membranes

#### 3.3.1. Effect of Plasticizer Concentration

Plasticizer is a component that forms a PIM membrane which acts as a ‘neutralizing’ polar polymer with its own polar group in order to reduce the distance between the polymer molecules and their intermolecular attractions. Furthermore, the presence of a plasticizer in the PIM membrane component produces a membrane that is strong, elastic, more stable and flexible. Figure 7 shows the phenol transport that occurs with variations in the concentration of the plasticizer added to the PIM membrane. Based on the results from the analysis using UV-vis spectrometry, it was found that the efficiency of plasticizer in transporting phenols under optimum conditions at a concentration of 0.3100 g in the receiving phase with phenol levels exceeding the required value was more than 40%.

The addition of plasticizer to the membrane increased the speed of transport and it was also possible to increase the diffusion of species that passes through the membrane. This addition also increases membrane elasticity, but may also be a barrier to transporting phenol to the receiving phase when the amount of plasticizer exceeds the required limit, beyond which a decrease in transport was attributed to intermolecular interaction between the target and plasticizer molecules, resulting in lower mass transfer [1,11].

#### 3.3.2. Effect of Salt Concentration

The presence of salts in the membrane has an important effect on membrane stability [22,23]. Membrane stability was predicted by looking at the effect of increasing the NaNO_3_ salt concentration. The addition of this compound was carried out with two variations, namely the addition of NaNO_3_ salt in the source and receiving phases.

The NaNO_3_ salt was added in the phenol transport process to determine the effect of salt concentration on the percentage of transport yield. Figure 8 shows a graph of the decrease in the percentage of phenol that was transported. Therefore, the greater the concentration of NaNO_3_ added to the source phase, the smaller the percentage of phenol that is transported to the receiving phase. Furthermore, the addition of salt in the receiving phase showed a decrease in the percentage of phenol that was transported using greater concentrations of NaNO_3_ salt.

#### 3.3.3. Lifetime PIM Membrane

The stability and resistance of the membrane were observed based on its ability to transport phenols at certain time intervals. Some works have reported the loss of membrane components as one of the main reasons for instability in the transport process of liquid membrane [24,25,26].

The lifetime of the membrane was also determined by measuring the pH value. It was found that when the pH value in the source phase was increased, the PIM became damaged. In addition, when the pH was at 9, the transport process also stopped, and it was concluded that the membrane had been damaged or had leaked.

Table 3 shows that the addition of salt in the source phase makes the PIM more stable in other to reduce membrane leakage. Salt with greater concentrations, such as 0.1 M NaNO_3_ and NaCl in the source phase makes the PIM have longer lifetime compared to samples without salt and lower concentrations.

The leakage rate of the components of membrane, namely PVC, DBE or co-EDVB 4%, with the use of NaNO_3_ 0.001 M was 21%. The other leakage data are shown in Table 4.

### 3.4. PIM Membrane Characterization

Figure 9 and Figure 10 show the characterization of the PIM membranes using FT-IR and SEM before and after transporting phenol. Figure 9 provides information that clearly shows the difference in the intensity of wavenumbers in the 3526 cm^−1^ region with low peaks, and in the current PIM membrane spectrum. There was also a shift in the region of the wavenumber 3391 cm^−1^. Furthermore, the absorption of this wavenumber shows the existence of stretching vibrations of O–H from hydrogen bonds, which causes a peak and shift towards the shorter wavenumbers with greater intensity.

Figure 10 shows the subsequent membrane characterization using SEM before (Figure 10a) and after transport (Figure 10b). The results show that the membrane had holes, and this was caused by the erosion of the membrane surface. Furthermore, the continuous use of membranes for transport enables its components to disappear, resulting in the surface becoming porous. This condition resulted in a ML loss value (membrane loss) which indicates missing membrane components. The occurrence of these losses within a certain time interval resulted in membrane leakage, thereby reducing the life span.

This mechanism was first mentioned by Danesi [27] and developed in detail by Takeuchi and Nakano [28]. According to this mechanism, during the operation of an SLM, both the interfacial tension and the contact angle might decrease with time because of the formation of metal complexes, contamination of the interfaces between the membrane and the aqueous solutions, dissociation of the chelating agents or other factors. When the interfacial tension decreases to a certain level, spontaneous emulsification would take place, causing the membrane liquid loss (ML loss) to the adjacent aqueous solutions. If the contact angle at the three-phase interface decreases to a critical value, the aqueous solution could penetrate the membrane pores. Based on this mechanism, the loss of ML should correspond to the compositions in the membrane phase. However, it is difficult for this mechanism to explain different losses of the carrier and the solvent into the aqueous solutions [24].

## 4. Conclusions

The membrane containing a PVC as a base polymer, 4% co-EDVB as a carrier compound, and DBE as a plasticizer can be used to transport the phenol at a pH of 5.5 in the source phase, the concentration of NaOH 0.1 M and phenol transport time of 72 h. The phenol transport kinetics using PIM membrane with 4% co-EDVB carrier compound following the order of reaction of order 1 had a k value of 1.19 × 10^−6^ m/s and a permeability value of 2.379 × 10^−8^ m/s. The addition of salt in the source phase was able to increase membrane lifetime up to 60 days. The loss of the components that made up the membrane during transport proved the presence of instability in the PIM membrane containing the carrier compound, 4% copoly-EDVB.

## Figures and Tables

**Figure 1 membranes-12-00295-f001:**
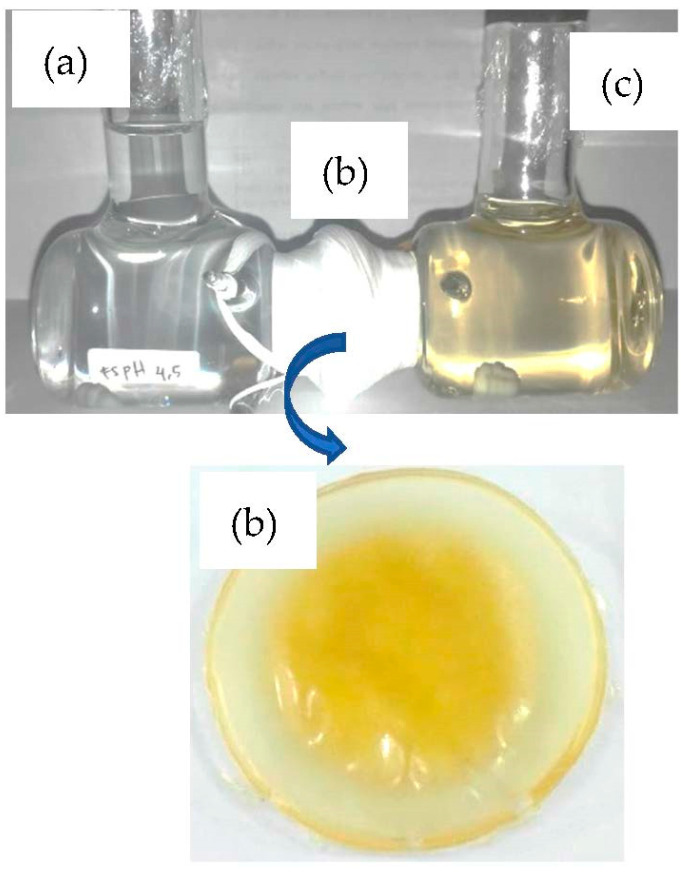
The PIMs installed between the two chambers: (**a**) source phase; (**b**) PIM membrane; (**c**) receiving phase.

**Figure 2 membranes-12-00295-f002:**
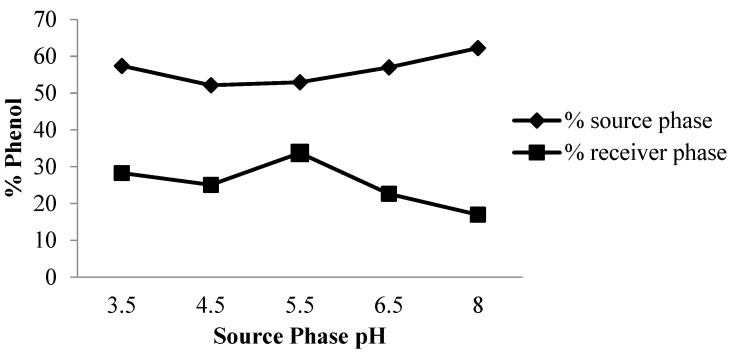
Curve effect of source phase pH on phenol levels.

**Figure 3 membranes-12-00295-f003:**
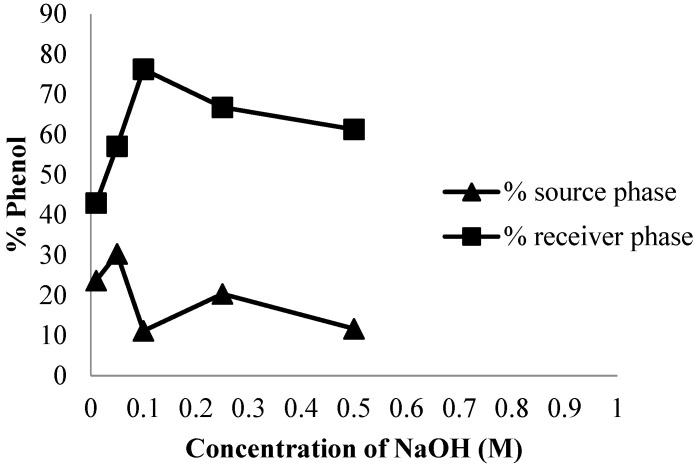
Phenol transport curves on the effect of NaOH concentration.

**Figure 4 membranes-12-00295-f004:**
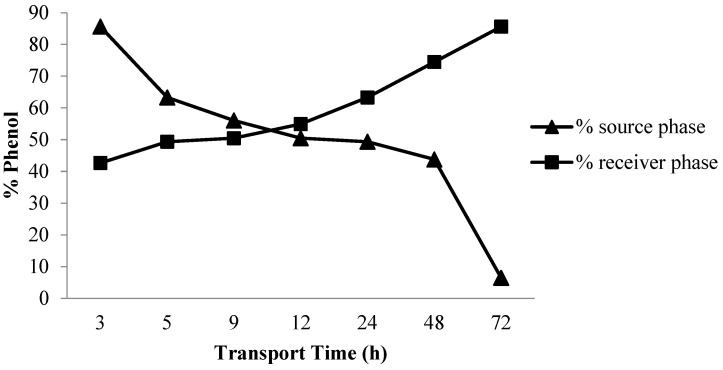
The phenol transport curve on the influence of time.

**Figure 5 membranes-12-00295-f005:**
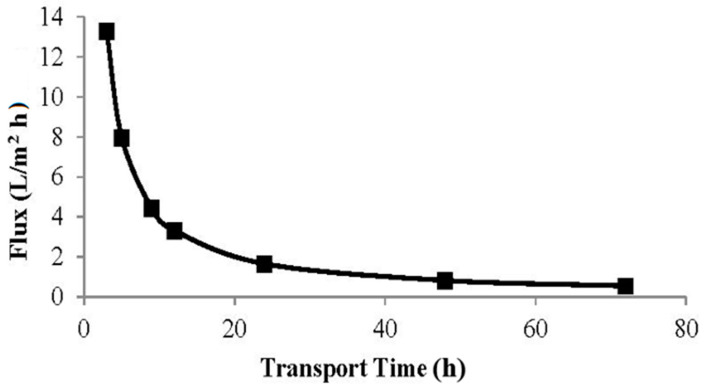
The curve of Flux value.

**Figure 6 membranes-12-00295-f006:**
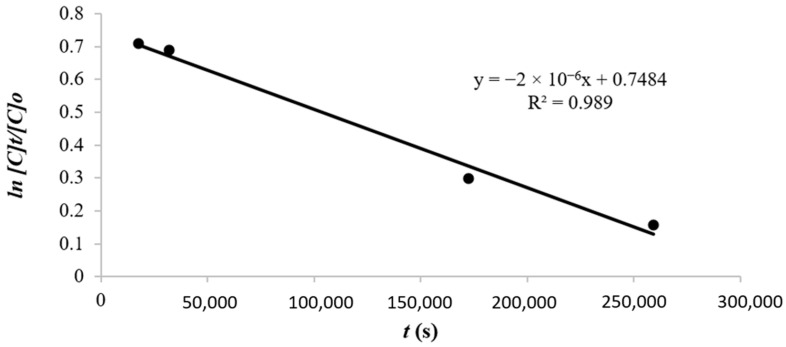
The curve of first-order phenol transport linear equation.

**Figure 7 membranes-12-00295-f007:**
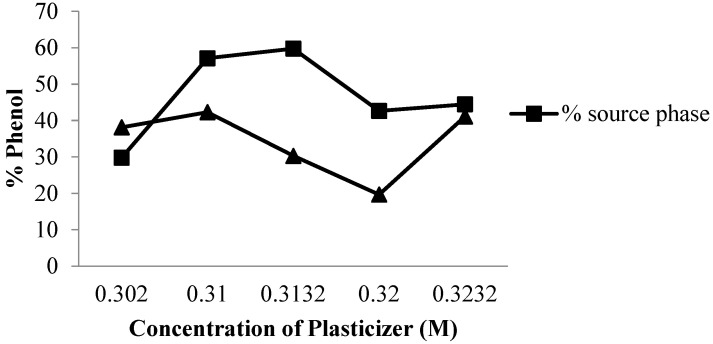
Curve effect of plasticizer concentration.

**Figure 8 membranes-12-00295-f008:**
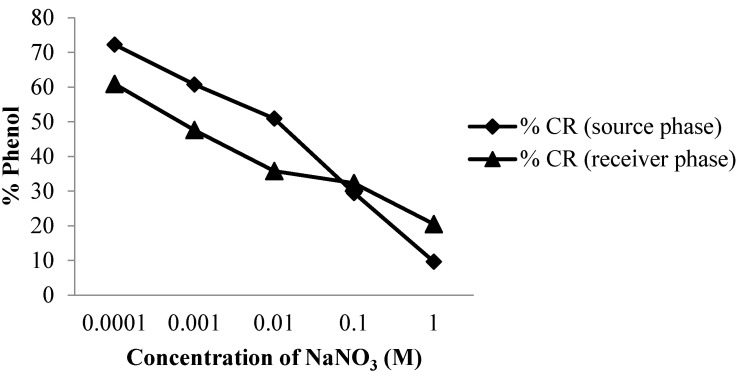
Graph of the effect of NaNO_3_ salt concentration in the source phases and the phenol in the receiver phase.

**Figure 9 membranes-12-00295-f009:**
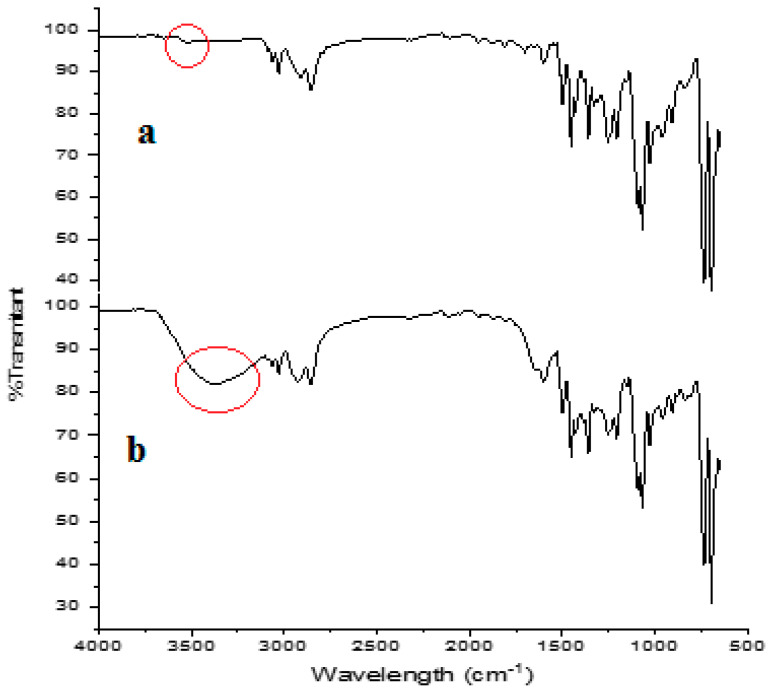
FT-IR spectra of PIM membrane (**a**) before transport and (**b**) after transport. Red cycle represents the hydroxyl peak on the PIM before and after transport.

**Figure 10 membranes-12-00295-f010:**
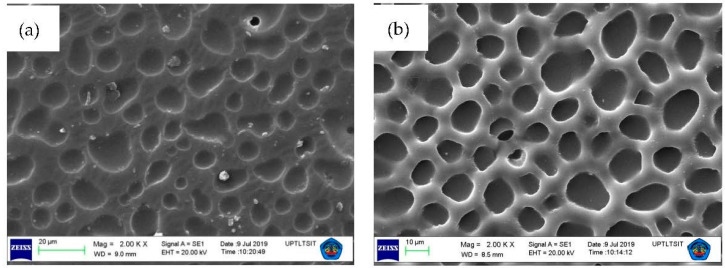
Characterization of SEM PIM membranes after transport with magnification of 2000× (**a**) before and (**b**) after transport.

**Table 1 membranes-12-00295-t001:** The flux value according to the variation in transport time.

Transport Time (h)	Flux Value (*J*) L/m^2^ h
3	13.27
5	7.96
9	4.42
12	3.31
24	1.65
48	0.83
72	0.553

**Table 2 membranes-12-00295-t002:** Values of first order and second order equations based on time variations.

*t* (h)	*t* (s)	[*C*]*_t_* ppm	[*C*]_0_ ppm	$ln\frac{{\left[C\right]}_{t}}{{\left[C\right]}_{0}}$
3	10,800	14.0	60	−0.851
5	18,000	24.0	60	−0.706
9	32,400	32.0	60	−0.6865
12	43,200	33.0	60	−0.599
24	86,400	37.0	60	−0.457
48	172,800	42.0	60	−0.295
72	259,200	43.0	60	−0.155

**Table 3 membranes-12-00295-t003:** Lifetime of PIM membranes.

Day	0	5	10	15	20	25	30	35	40	45	52	60	>60
[Salt]	pH												
0 M	5.50	6.5	6.87	7.27	7.86	8.57	8.90	9.27	-	-	-	-	-
NaNO_3_ 0.01 M	5.50	6.35	6.86	7.52	7.62	8.07	8.45	8.62	8.77	8.89	8.9	9.2	-
NaCl 0.01 M	5.50	5.62	7.44	7.50	7.71	7.90	8.37	8.75	8.9	9.1	-	-	-
NaNO_3_ 0.1 M	5.50	6.59	7.32	7.52	7.74	7.81	7.88	8.12	8.3	8.5	8.6	8.8	9.0
NaCl 0.1 M	5.50	5.68	7.15	7.32	7.45	7.58	7.70	7.91	8.0	8.2	8.3	8.5	8.9

**Table 4 membranes-12-00295-t004:** Membrane liquid (ML) loss in the presence of NaNO_3_ in the receiving phase.

[NaNO_3_] M	PIM Membrane			
	Mass (g)		ML Loss (g)	ML Loss(%)
	Before	After		
0.001 M	0.4595	0.365	0.0945	21
0.01 M	0.4765	0.382	0.0945	20
0.1 M	0.4649	0.382	0.0829	18
1 M	0.432	0.371	0.061	14

## Data Availability

The data produced from this study can be requested from the corresponding author, upon reasonable request.

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
