# Peer review of "Optimization and Evaluation of Polymer Inclusion Membranes Based on PVC Containing Copoly-EDVB 4% as a Carrier for the Removal of Phenol Solutions"

_membranes, 2022, doi:10.3390/membranes12030295_

Round 1

Reviewer 1 Report

 Article 

Optimization and Evaluation of Polymer Inclusion Membranes  based on PVC containing Copoly-EDVB 4% as a Carrier for Re-3 moval of Phenol Solutions

The comments

The abstract should be more attractive, and the novelty of the work should be highlighted.

There are many English mistakes, which should be corrected.

The results should be explained and the discussion section should be improved.

The conclusion part needs to be revised

  1. There are some sentences that don't appear very clear without the word "respectively", i. e. the lines: 12-13, 59-60, 77-78
  2. Line 58: The plural verb "include" does not appear to agree with the singular subject "This". Consider changing the verb form for subject-verb agreement.
  3. Lines 70, 123, 242: It is necessary to keep the same writing of the units (g or grams).
  4. Lines 79-82; 88-91; 98-100; 127-129; 136-138; 144-146: This sentence "Afterward, the phenol concentrations contained in the source and receiving phases were 89 analyzed using a UV-Vis spectrophotometer at a maximum wavelength of 456nm and the 90 4-aminoantiphyrine method.", is repeated many times, just state it in the first part. Similarly for "50 mL of NaOH and 60 ppm phenol was added to the receiving and source phases of each chamber", in several lines, i.e. 59-60, 77-78, 85-86, and "The PIMs were installed between the two chambers for easy transportation" in lines 76-77, 84-85 and 94-95.
  5. Lines 101-105; 204-209: There is no need to repeat the same paragraph.
  6. Line 119: The points in equation 3 must be removed.
  7. Line 170: An incorrect form of the verb "be". Consider changing it to "been".
  8. Line 182: It's about the figure "2" not "3".
  9. Lines 284, 286, 288: The word "wave numbers" seems to be miswritten. Consider changing it to "wavenumbers".
  10. Lines 194-195; 199-200: It is not necessary to repeat the same interpretation in the same part "This is because the interactions that occur between phenols in the membrane and receiving phases increased along with the time needed for transport".
  11. Line 211: Try to make the legend appropriate to the representation.
  12. Line 259: In the figure 7, the word "Phase" is repeated twice (Receiver Phase Phase).
  13. Line 292: Consider correcting the legend of Figure 9, and the appointment (a, b) must be indicated on the SEM images.
  14. Lines 303, 304: The word "phenol" is miswritten. Also, it seems that there is an article usage problem" in source phase". Consider putting the article "the".
  15. Line 307: It seems that the preposition use may be incorrect in "to increase of membrane lifetime". Consider deleting the preposition "of".

Author Response

We would like to thank you for giving us very excellent comments to our manuscript. We have attempted to correction as necessary.  All corrections are highlight in yellow.

  1. In the abstract the highlight of novelty of this manuscript has been added and highlighted at the end of the abstract.
  2. The English mistakes have been attempted to be corrected, however, upon acceptance of this manuscript, we will be happy to another proofreading in order to make the English much smoother.
  3. The discussion has been improved as requested.
  4. There are some sentences that don't appear very clear without the word "respectively", i. e. the lines: 12-13, 59-60, 77-78, have been adjusted accordingly.
  5. The repetitions of some sentences have been minimized.

Response to the Comments and Suggestions for Authors

 Article 

Optimization and Evaluation of Polymer Inclusion Membranes  based on PVC containing Copoly-EDVB 4% as a Carrier for Removal of Phenol Solutions

The comments

The abstract should be more attractive, and the novelty of the work should be highlighted.

Novelty: Lifetime membrane bisa menjadi 60 hari dengan penambahan garam NaNO3 atau NaCl 0.1 M pada fasa sumber.

There are many English mistakes, which should be corrected.

The results should be explained and the discussion section should be improved.

The conclusion part needs to be revised

  1. There are some sentences that don't appear very clear without the word "respectively", i. e. the lines: 12-13, 59-60, 77-78

It has been added

  1. Line 58: The plural verb "include" does not appear to agree with the singular subject "This". Consider changing the verb form for subject-verb agreement.

It has been adjusted.

  1. Lines 70, 123, 242: It is necessary to keep the same writing of the units (g or grams).

It has been fixed.

  1. Lines 79-82; 88-91; 98-100; 127-129; 136-138; 144-146: This sentence "Afterward, the phenol concentrations contained in the source and receiving phases were 89 analyzed using a UV-Vis spectrophotometer at a maximum wavelength of 456nm and the 90 4-aminoantiphyrine method.", is repeated many times, just state it in the first part. Similarly for "50 mL of NaOH and 60 ppm phenol was added to the receiving and source phases of each chamber", in several lines, i.e. 59-6077-7885-86, and "The PIMs were installed between the two chambers for easy transportation" in lines 76-7784-85 and 94-95.

They have been fixed.

  1. Lines 101-105; 204-209: There is no need to repeat the same paragraph.

It has been fixed.

  1. Line 119: The points in equation 3 must be removed.

It has been fixed

  1. Line 170: An incorrect form of the verb "be". Consider changing it to "been".

It has been fixed.

  1. Line 182: It's about the figure "2" not "3".

Yes, it has been fixed.

  1. Lines 284, 286, 288: The word "wave numbers" seems to be miswritten. Consider changing it to "wavenumbers".

Yes, they have been fixed.

  1. Lines 194-195; 199-200: It is not necessary to repeat the same interpretation in the same part "This is because the interactions that occur between phenols in the membrane and receiving phases increased along with the time needed for transport".

It has been fixed.

  1. Line 211: Try to make the legend appropriate to the representation.

In our opinion, the legend in Figure 3 has been very distinctive, We are confused how can we make them clearer.

  1. Line 259: In the figure 7, the word "Phase" is repeated twice (Receiver Phase Phase).

It has been deleted

  1. Line 292: Consider correcting the legend of Figure 9, and the appointment (a, b) must be indicated on the SEM images.

The appointment has been added

  1. Lines 303, 304: The word "phenol" is miswritten. Also, it seems that there is an article usage problem" in source phase". Consider putting the article "the".

They have been fixed

  1. Line 307: It seems that the preposition use may be incorrect in "to increase of membrane lifetime". Consider deleting the preposition "of".

It has been corrected

Reviewer 2 Report

I would suggest the authors to strengthen the manuscript by revising the manuscript based on the following comments.

Abstract:

Line 14: What is co-EDVB mean? Please, indicate the meaning of abbreviations before you use them for the first time.

Line 18: Please, change “1M” by “1 M”. The International System of Units (SI) recommends inserting a space between a number and a unit of measurement units, please use “1 M” instead “1M”.  You must use this for the rest of document.

Introduction:

In general, introduction is too short. It can be longer and modified, because some information about fundamental issues as general used or preparation of PIM are missing. In addition, some information about the physico-chemical properties of formed link (carrier-phenol) must been included.

Line 24: The other conventional methods must been defined, and the advantages of using the proposed method should be explained. This must also be detailed for the removal of phenol.

Materials and methods:

Line 58: In order to understand the device, a scheme and an image of it must be appropriate

Line 60: Please, indicate the procedure and reason for sterilization

Lines 63-66: Please indicate the model, commercial name, city and country of each device

Line 70: The compositions of PIM must been indicated more details, Do they used a total weight of 0.270 PVC, 0.5400 EDVB , and 1.0800 g DBE? How did authors arrive to conclusion this composition?

Data on the size and shape of the membrane used should also be included.

Line 71: Please, clarify what the relationship 10:32:58 means

Line 81: the 4-aminoantiphyrine method´s or its reference must be indicated.

Line 98: Indicate if a sample volume was extracted at each time, what volume was taken, if the volume was replaced, and if the final concentration was corrected based on the extracted volume.

Ecuantion 1 and 2:  each term in these equations must been defined

ine 122 and 123: the author 0.3032, 0.3100, 122 0.3132, 0.3200, 0.3232 grams for plasticizer concentrations,  I would recommend an explanation for this curious and similar weights.

Line 127: clarify whether the previous experiments are carried out under conditions of not exposed to sunlight. If not, indicate why it is done in 2.4

Results:

It is strange that firstly authors did not optimization of membrane composition and then experimental conditions of measurements step by step and not simultaneously.

Authors must indicate the number of repetitions of each experiment (n) and the +/-.  This is very important, if no repetitions have been made, they must be included in the work before being published. In addition, application of proper quality assurance/quality control (QA/QC) procedures is vital for the measurement results to be treated as a source of reliable analytical information. Consequently, I suggest that a separate section devoted to QA/QC be added to the manuscript.  Please provide such information

Finally, for comparison, I would recommend to add an error bar also to the graphs

Line 272: the author wrote:  “It was found that when the pH value in the source phase was increased, the PIM became damaged”. I would recommend an explanation for this pH increased with the time. how can it really be usable? I would have expected a broad discussion on this in the next paragraph.

It should be also tested how The PIM can be applied to real samples and the proposed procedure for masking of possible interference should be verified.   

Author Response

We would like to thank you for giving us very excellent comments to our manuscript. We have attempted to correction as necessary.  All corrections are highlight in yellow.

Response to the Comments and Suggestions for Authors

I would suggest the authors to strengthen the manuscript by revising the manuscript based on the following comments.

Abstract:

Line 14: What is co-EDVB mean? Please, indicate the meaning of abbreviations before you use them for the first time.

It has been provided

Line 18: Please, change “1M” by “1 M”. The International System of Units (SI) recommends inserting a space between a number and a unit of measurement units, please use “1 M” instead “1M”.  You must use this for the rest of document.

 All concentration of M have been fixed

Introduction:

In general, introduction is too short. It can be longer and modified, because some information about fundamental issues as general used or preparation of PIM are missing. In addition, some information about the physico-chemical properties of formed link (carrier-phenol) must been included.

We have added description about how PIM is prepared, we have also added the physico-chemical characteristic of carrier compound in the introduction.

Line 24: The other conventional methods must been defined, and the advantages of using the proposed method should be explained. This must also be detailed for the removal of phenol.

We have added some conventional methods

Materials and methods:

Line 58: In order to understand the device, a scheme and an image of it must be appropriate

We have provided the setting of PIM in Figure 1.

Line 60: Please, indicate the procedure and reason for sterilization

The sterilization is done by washing the membrane, followed with heating it in a hot water bath and oven it at 100 °C

Lines 63-66: Please indicate the model, commercial name, city and country of each device

We have provided the details of each product used.

Line 70: The compositions of PIM must been indicated more details, Do they used a total weight of 0.270 g PVC, 0.5400 g co-EDVB , and 1.0800 g DBE? How did authors arrive to conclusion this composition?

PIM membrane consists of these three materials. PIM membranes were fabricated using the protocol developed by Nghiem et al. [1]

Data on the size and shape of the membrane used should also be included.

Line 71: Please, clarify what the relationship 10:32:58 means

This is the comparison of composition of the membrane used and we only used the composition available in the literature [1]

Line 81: the 4-aminoantiphyrine method´s or its reference must be indicated.

It has been provdided

Line 98: Indicate if a sample volume was extracted at each time, what volume was taken, if the volume was replaced, and if the final concentration was corrected based on the extracted volume.

In the Effect of time, we used a different chamber for each time of experiment. So we used chambers based on the number of experiments carried out.  

Ecuantion 1 and 2:  each term in these equations must been defined

We have provide each definition for each term.

Line 122 and 123: the author 0.3032, 0.3100, 122 0.3132, 0.3200, 0.3232 grams for plasticizer concentrations,  I would recommend an explanation for this curious and similar weights.

Why we did that addition with those numbers are based on the explanation: According to Nghiem et al. [1], plasticizers are very influential on the flexibility/elongation of the membrane, if too much will be oily, if too little will be stiff, sensitive to the weight ratio of the plasticizer.

Line 127: clarify whether the previous experiments are carried out under conditions of not exposed to sunlight. If not, indicate why it is done in 2.4.

Previous experiments were also done under condition of not exposed to sunlight, as phenol will melt upon expose to sunlight

Results:

It is strange that firstly authors did not optimization of membrane composition and then experimental conditions of measurements step by step and not simultaneously.

Membrane compositions are followed the method developed by Nghiem et al. [1]. We focus on the parameters that affect the ability of phenol transport by the membrane

Authors must indicate the number of repetitions of each experiment (n) and the +/-.  This is very important, if no repetitions have been made, they must be included in the work before being published. In addition, application of proper quality assurance/quality control (QA/QC) procedures is vital for the measurement results to be treated as a source of reliable analytical information. Consequently, I suggest that a separate section devoted to QA/QC be added to the manuscript.  Please provide such information

The repetitions were only carried out on the phenol analysis. We can not provide QA/QC.

Finally, for comparison, I would recommend to add an error bar also to the graphs

Unfortunately, we are unable to provide error bar as we did not do the repetition of phenol transport, rather we did repeat the analysis the phenol concentration. 

Line 272: the author wrote:  “It was found that when the pH value in the source phase was increased, the PIM became damaged”. I would recommend an explanation for this pH increased with the time. how can it really be usable? I would have expected a broad discussion on this in the next paragraph.

The pH of the source phase is initially acidic, when the transport process occurs then the pH of the source phase rises, this indicates that there is NaOH that permeates/moves/diffuses from the receiving phase to the source phase so that the pH in the source phase is increase. The increase of the pH is an indication that the PIM has leaked.

It should be also tested how The PIM can be applied to real samples and the proposed procedure for masking of possible interference should be verified.   

Implementation of phenol processing (removal and recovery), adapted to the condition "how the company processes its waste processing". But at least, that the waste stream containing phenol must pass through the reactor where the rector has been installed (set) complete with PIM membrane. Furthermore, the waste stream containing phenol can be recovered and/or removed so that the wastewater coming out of the reactor is clean of phenol pollutants.

Reviewer 3 Report

(1). In lines 107-119, the ordinals of the equations are not aligned, and there are many points in front of equation 3, but not in equations 1 and 2. F in equation 1 is not explained.

(2). In line 182, "Based on figure 3, the optimum phenol transport was 182 obtained with a pH of 5.5 at 0.1 M NaOH concentration". It should be based on Figure 2, not Figure 3.

(3) In Section 3.2.2. The effect of salt concentration, why NaNO3 was used in the experiment, rather than NaCl or KNO3 on the effect of salt concentration? Or other common salt effects, sodium nitrate. What's the point?

(4). In line 299, "The occurrence of these losses within a certain time in-299 terval resulted in membrane leakage, thereby reducing the life span." The authors propose that the membrane will be lost within a certain period of time, resulting in membrane leakage. What is the leakage rate?

(5) Because of the adsorption of phenol in an aqueous solution. Therefore, does the material contact angle between the modified film and the unmodified film change? There's no explanation here.

6). Finally, your English grammar is wrong in tenses. I hope you can check and correct it carefully.

Author Response

We would like to thank you for giving us very excellent comments to our manuscript. We have attempted to correction as necessary.  All corrections are highlight in yellow.

Response to Comments and Suggestions for Authors

(1). In lines 107-119, the ordinals of the equations are not aligned, and there are many points in front of equation 3, but not in equations 1 and 2. F in equation 1 is not explained.

We have corrected and added definition of each term used.

(2). In line 182, "Based on figure 3, the optimum phenol transport was 182 obtained with a pH of 5.5 at 0.1 M NaOH concentration". It should be based on Figure 2, not Figure 3.

Yes we have corrected accordingly.

(3) In Section 3.2.2. The effect of salt concentration, why NaNO3 was used in the experiment, rather than NaCl or KNO3 on the effect of salt concentration? Or other common salt effects, sodium nitrate. What's the point?

It is a series of experiments. In this work we use NaNO3. In other works we are currently doing we are also trying to use NaCl and KNO3. Once this experiment finished, we will try to write another manuscript and we will compare it with the data reported in this manuscript. The experiments are still running and involving some students in our laboratory.

(4). In line 299, "The occurrence of these losses within a certain time interval resulted in membrane leakage, thereby reducing the life span." The authors propose that the membrane will be lost within a certain period of time, resulting in membrane leakage. What is the leakage rate?

We have added a Table discussing about the leakage rate in the presence of NaNO3 in various concentration

(5) Because of the adsorption of phenol in an aqueous solution. Therefore, does the material contact angle between the modified film and the unmodified film change? There's no explanation here.

We have added the discussion about this contact angle.

6). Finally, your English grammar is wrong in tenses. I hope you can check and correct it carefully.

We have actually done proofreading with native, and we have tried our best to improve the English. Upon acceptance of the manuscript, I will do another proofreading to meet the standard of English used.

Round 2

Reviewer 1 Report

- Figure 5: Flux instead of Fluks on the y-axis
- (Page 6, line 229) Flux is the number of samples, this definition is false.

- Figure 6 and not figure 65
- Figure 6, it is more convenient to present ln(C0/Ct) to have a positive slope;
- For an order 1, the speed constant k is in (time-1) and not in (m/s)
- Figur 10, it is better to compare two images with the same magnification
- Explain the assay method
- It is necessary to improve the conclusion

Author Response

Response to Reviewer 1

- Figure 6 and not figure 65

It has been corrected.
- Figure 6, it is more convenient to present ln(C0/Ct) to have a positive slope;

The slope was negative as the flux value was decrease with the increase of transport time

- For an order 1, the speed constant k is in (time-1) and not in (m/s)

k in our work is coefficient of mass transfer, thus the unit is m/s.

- Figur 10, it is better to compare two images with the same magnification

We have replaced with the correct SEM image with the same magnification

Before transport

After transport

- Explain the assay method

The following is the essay method using 4-aminoantiphyrine method and it has been added in the procedure in the manuscript.

5 mL of sample in the source phase, receiving phase and  the phenol standard  with several concentration variations were added with 5 mL of double distilled water so that the final volume was 10 mL. The pH of the solutions in the source phase and phenol standard was adjusted to 10 ± 0.2 using 1M NH4OH, phosphate buffer, while pH for the receiving phase was adjusted with 0.5 M HCl, then 1mL of 4-aminoantipyrine 2% and 8% potassium ferricyanide were added. The solution was allowed to stand until the color of solution changed to pink.

After the color change occurred, the solution was transferred into a separating funnel and added with 5 mL of chloroform. The separating funnel was shaken and allowed to stand for a while until separation occurred, then the organic layer or chloroform layer (bottom) was separated. The chloroform extract obtained was measured for absorbance using a UV-vis spectrophotometer at a wavelength (λ) of 456 nm.

- It is necessary to improve the conclusion

We have improved the conclusion by adding a statement.

Reviewer 2 Report

see attached document

Author Response

Response to Reviewer 2

Although the quality and presentation of the work has improved, I consider that its publication can only be recommended after mayor revision. The authors must carry out repetition of the experiments that allow establishing that the results they present are reliable and the conclusions are well-founded.

Below I detail aspects that I consider not solved by the authors with respect to the first versión (in red)

Materials and methods:

-Line 60: Please, indicate the procedure and reason for sterilization

The sterilization is done by washing the membrane, followed with heating it in a hot water bath and oven it at 100 °C

The author had indicated the procedure but not the reason for sterilization

The sterilization here meant the treatment of all glassware used by researchers by washing glassware with soapy water and rinsing thoroughly with water, after that all containers are soaked with 5% HNO3 for 24 hours which serves to remove contamination and then dry using oven at 100oC, then storage is carried out in a closed state until the glassware is used.

-Line 70: The compositions of PIM must been indicated more details, Do they used a total weight of 0.270 g PVC, 0.5400 g co-EDVB , and 1.0800 g DBE? How did authors arrive to conclusion this composition?

PIM membrane consists of these three materials. PIM membranes were fabricated using the protocol developed by Nghiem et al. [1]

It must include in more detail how this proportion has been calculated, in addition to the reference in the text of section 2.2

The composition of the PIM membrane is made up of 3 components, namely carrier compound, PVC (as a supporting membrane) and DBE (as a plasticizer). Because the components that make up the membrane must consist of these three components, the use of plasticizers is designed in such a way that the membrane formed is neither too rigid nor too elastic, although in the prepration of the membrane also follows the procedure developed by Nghiem et al. [1]

Data on the size and shape of the membrane used should also be included.

The composition of PIM membrane components

Type of

Membrane

Carrier compound

(g)

PVC

(g)

DBE

(g)

Total weight

(g)

T27

0.027

0.0864

0.1566

0.2700

T54

0.054

0.1728

0.3132

0.5400

T108

0.108

0.3456

0.6264

1.0800

-Line 71: Please, clarify what the relationship 10:32:58 means

This is the comparison of composition of the membrane used and we only used the composition available in the literature [1]

Clarify if it is a weight ratio, molar, etc., indicate the units. In my opinion, It remains unclear how to get there and what it means

The PIM membrane was molded or made with a total weight of T27: 0.270, T54: 0.5400, and & T108: 1.0800 g in a mold equipped with a magnetic stirrer.

The ratio of co-EDVB as a carrier compound, PVC as a basic polymer, and DBE as a plasticizer is 10:32:58 with a PIM membrane composition as shown in the table above.

 -Line 81: the 4-aminoantiphyrine method´s or its reference must be indicated.

It has been provided  in detail in the procedure

The reference that has been included ([12] also does not indicate what method it is and how it is carried out. Include a valid reference or explain the measurement procedure.

 -Ecuantion 1, 2 and 3:

Ecuation 2: x must been deleted, use · or an space between A and t

Ecuation 3: k must been defined

-Line 122 and 123: the author 0.3032, 0.3100, 122 0.3132, 0.3200, 0.3232 grams for plasticizer concentrations, I would recommend an explanation for this curious and similar weights.

Why we did that addition with those numbers are based on the explanation: According to Nghiem et al. [1], plasticizers are very influential on the flexibility/elongation of the membrane, if too much will be oily, if too little will be stiff, sensitive to the weight ratio of the plasticizer.

Please include this clarification in the text of the article

It was clearly mentioned in reference [1] point 4.2 about Plasticizer concentration and they were explaining about the effect of plasticizer completely, and we have added some statements in the manuscript.

-Line 127: clarify whether the previous experiments are carried out under conditions of not exposed to sunlight. If not, indicate why it is done in 2.4.

Previous experiments were also done under condition of not exposed to sunlight, as phenol will melt upon expose to sunlight

Please include this clarification in the text of the article

Based on the chemical data sheet that safe storage conditions are in a cool place, therefore the same treatment during the study was carried out at room temperature and not exposed to sunlight.

Results:

-Authors must indicate the number of repetitions of each experiment (n) and the +/-. This is very important, if no repetitions have been made, they must be included in the work before being published. In addition, application of proper quality assurance/quality control (QA/QC) procedures is vital for the measurement results to be treated as a source of reliable analytical information. Consequently, I suggest that a separate section devoted to QA/QC be added to the manuscript. Please provide such information

The repetitions were only carried out on the phenol analysis.

We can not provide QA/QC.

This section is very important. The reliability of a result must be based on the absence of random or systematic errors, to avoid them it is essential to carry out replicas of each of the experiments carried out. If no repetitions have been made, they must be included in the work before being published with the proper quality assurance/quality control (QA/QC) procedures.

This work has been concluded sometimes ago, thus it is impossible to add the repetition data as requested by the reviewer.

For your information, we have previously published a similar work (the difference is only the carrier compound) as in reference no. [14] and we did not provide QA/QC data.
